# Chimeric symbionts expressing a *Wolbachia* protein stimulate mosquito immunity and inhibit filarial parasite development

Sara Epis [1,2,10], Ilaria Varotto-Boccazzi [1,2,10], Elena Crotti[3], Claudia Damiani[2,4], Laura Giovati [5], Mauro Mandrioli [6], Marco Biggiogera [7], Paolo Gabrieli[1,2], Marco Genchi[8], Luciano Polonelli[5], Daniele Daffonchio [9], Guido Favia[2,4] & Claudio Bandi [1,2✉]

*Wolbachia* can reduce the capability of mosquitoes to transmit infectious diseases to humans and is currently exploited in campaigns for the control of arboviruses, like dengue and Zika. Under the assumption that *Wolbachia*-mediated activation of insect immunity plays a role in the reduction of mosquito vectorial capacity, we focused our attention on the *Wolbachia* surface protein (WSP), a potential inductor of innate immunity. We hypothesized that the heterologous expression of this protein in gut- and tissue-associated symbionts may reduce parasite transmission. We thus engineered the mosquito bacterial symbiont *Asaia* to express WSP (*Asaia*^WSP^). *Asaia*^WSP^ induced activation of the host immune response in *Aedes aegypti* and *Anopheles stephensi* mosquitoes, and inhibited the development of the heartworm parasite *Dirofilaria immitis* in *Ae. aegypti*. These results consolidate previous evidence on the immune-stimulating property of WSP and make *Asaia*^WSP^ worth of further investigations as a potential tool for the control of mosquito-borne diseases.

---

[1] Department of Biosciences and Pediatric Clinical Research Center "Romeo and Enrica Invernizzi", University of Milan, Milan, Italy. [2] Centro Interuniversitario di Ricerca sulla Malaria/Italian Malaria Network, Milan, Italy. [3] Department of Food, Environmental and Nutritional Sciences, University of Milan, Milan, Italy. [4] School of Biosciences and Veterinary Medicine, University of Camerino, Camerino, Italy. [5] Department of Medicine and Surgery, University of Parma, Parma, Italy. [6] Department of Life Sciences, University of Modena and Reggio Emilia, Modena, Italy. [7] Department of Biology and Biotechnology "L. Spallanzani", University of Pavia, Pavia, Italy. [8] Department of Veterinary Sciences, University of Parma, Parma, Italy. [9] King Abdullah University of Science and Technology, Red Sea Research Center, Thuwal, Saudi Arabia. [10] These authors contributed equally: Sara Epis, Ilaria Varotto-Boccazzi. ✉email: claudio.bandi@unimi.it

The microbial communities of insects and mites of medical relevance, such as mosquitoes, sandflies and ticks, have attracted a great deal of attention, and it is now well established that arthropod-associated microbes influence the fitness of the arthropod hosts, as well as their capability to transmit pathogens to humans and animals[1]. Mosquitoes have been in the focus of this research area, with over 150 papers published in the last 5 years, on their microbiota and, accordingly, on their symbionts. Two symbiotic bacteria found in mosquitoes have emerged for their prominent biological role in these insects, as well for their potential utility for the control of mosquito-borne diseases: *Asaia* spp. and *Wolbachia pipientis*. Representatives of the genus *Asaia* have been detected in different mosquito species; more in general, they have been observed in several insects[2,3]. *Asaia* spp. are extracellular acetic acid bacteria, which can easily be cultured in cell-free media and have already been engineered at both the plasmid and chromosomal level, also for the expression of molecules interfering with the development of malaria parasites[2,4–6]. These bacteria colonize the gut, salivary glands and reproductive organs of both male and female mosquitoes. From the reproductive organs, *Asaia* can be transmitted venereally form males to females and vertically from mother to offspring, via egg-smearing[7]. From the salivary glands, *Asaia* can be transmitted horizontally among adults through cofeeding[4,7,8]. The actual capability of *Asaia* to spread into mosquito populations has recently been demonstrated in semi-field conditions[9]. Based on the above characteristics, *Asaia* bacteria have been defined as very promising mosquito symbionts, suitable for the control of vector-borne diseases through paratransgenesis[6]. In vector-borne disease control, paratransgenesis is the use of microbial symbionts manipulated for the expression of molecules that determine, either directly or indirectly, the reduction of pathogen transmission[10,11].

The intracellular bacterium *Wolbachia* is probably the most widespread intracellular symbiont in arthropods[12], found also in filarial nematodes[13], and already used in the field for the control of mosquito-borne viruses[14]. Indeed, through alteration of fatty acid intracellular trafficking, competition for cholesterol, manipulation of miRNAs expression and/or upregulation of innate immunity responses, *Wolbachia* strains have been shown to interfere with the transmission of human pathogens by mosquitoes (e.g. dengue and Zika viruses, malaria parasites and filarial worms[15–20]). However, the biological effects of *Wolbachia* infection on the insect host and its vector competence are not predictable; for example, Dodson and co-workers reported that *Wolbachia* enhances West Nile viral infection in the mosquito *Culex tarsalis*[21]. Field applications for the control of dengue virus transmission through the release of *Wolbachia*-infected *Aedes aegypti* mosquitoes have been established since 2011, with very effective results[22,23]. The exploitation of *Wolbachia* in paratransgenesis is however impaired by the characteristics of this bacterium: it is an obligate intracellular symbiont and it is not culturable in cell-free media, and thus not easy to be engineered[24].

An alternative approach to exploit *Wolbachia* could be the identification of molecules from this bacterium able to stimulate the immune system of the mosquito, thus potentially interfering with the insect vectorial capacity. The major surface protein (WSP) of the *Wolbachia* hosted by the nematode *Dirofilaria immitis* has been shown to induce an upregulation of immune gene transcription in cells from the mosquito *Anopheles gambiae*[25], which is normally not infected by *Wolbachia* (except for some local populations[26]). WSP has also been shown to activate innate immune responses in mammalian models, supporting the activity of this protein as a general trigger of innate immune activation both in insects and in mammals[27].

According to the above evidence and assumptions, we aimed to combine properties of *Asaia* and *Wolbachia* symbionts, in order to confer an increased immune-activating capability, derived from *Wolbachia*, to the culturable *Asaia* of mosquitoes. To accomplish this aim, we engineered *Asaia* SF2.1 strain[4] for the expression of WSP from the *Wolbachia* infecting the nematode *D. immitis*[25,27]. We then tested the capability of the modified bacterium to colonize mosquito organs, to stimulate the immune system, to induce phagocytosis and to interfere with the development of filarial parasites.

## Results

**WSP expression by *Asaia* SF2.1 and fitness of the bacteria.** A schematic presentation of the *Asaia*-pHM4-WSP (hereafter *Asaia*[WSP]) construct is shown in Supplementary Fig. 1a, b. Plasmid pHM4-WSP was constructed by inserting the *wsp* gene cassette flanked by *Not*I sites in the plasmid pHM4. An E-tag epitope was included for immunodetection purposes; the production of WSP protein by *Escherichia coli* and *Asaia* sp. was evaluated by Western-blot and immunofluorescence assays, with anti-E-tag antibodies. As shown in Supplementary Fig. 1c, *Asaia*[WSP] is able to express the protein (26 kDa), while, as expected, *Asaia*-pHM4 (hereafter *Asaia*[pHM4]) does not produce the WSP protein (the same results were observed for *E. coli*). The expression of the *wsp* gene was also verified by RT-qPCR using bacteria grown at different optical densities (ODs) (Supplementary Fig. 1d): no expression was observed for *Asaia*[pHM4], while *Asaia*[WSP] expressed the *wsp* gene, with a substantial increase of the expression from OD 0.5 ($6.253 \pm 0.385$) to OD 1 ($9.970 \pm 0.391$). Based on these results, we decided to use OD 1 for other analyses. In addition to Western blot analysis (see above and Supplementary Fig. 1c), the expression/production of WSP protein was also verified by immunodetection: both immunofluorescence (Supplementary Fig. 2a–d) and immunogold staining (Supplementary Fig. 1e–g) confirmed the production of the protein by *Asaia*[WSP] bacteria, while no staining (or a very faint background) was observed in *Asaia*[pHM4] control bacteria. The anti-Etag immunogold staining on *Asaia*[WSP] revealed a pattern of colloidal gold deposits associated with the bacterial cells (Supplementary Fig. 1e,f). To verify if the production of the heterologous WSP had negative effects on *Asaia* growth, we analyzed growth curves of the bacteria at different pH conditions along a 24-h period. This test was also performed to reproduce the different pH condition in mosquito organs and to test the capability of *Asaia* to survive and grow. In this fitness assay, we compared the strain *Asaia*[wt] with the engineered strains carrying the plasmids pHM4 (*Asaia*[pHM4]) or *Asaia*[WSP]: the mean maximal growth rates (MGRs) of wild type and the two transformed strains were not significantly different in almost all the tested growth conditions, with the exception of the MGRs of *Asaia*[wt] and *Asaia*[WSP] at pH 4 ($p = 0.038$) (Fig. 1). In conclusion, WSP expression does not significantly affect the fitness of *Asaia*[wt] in most of the tested pH conditions.

**In vitro phagocytosis test and immune-related gene expression.** Phagocytosis tests on haemocytes from *Ae. aegypti* and *An. stephensi* revealed significant differences, after the stimulation with *Asaia*[pHM4] or *Asaia*[WSP] for 1 ($p < 0.0001$ and $p = 0.0089$, respectively) and 2 h ($p = 0.0001$ and $p < 0.0001$) (Fig. 2a,b). The expression of the two selected antimicrobial peptides, defensin and cecropin, and the nitric oxide synthase (*NOS*) was investigated on haemocytes from *An. stephensi* and *Ae. aegypti*, after an in vitro stimulation with the two engineered bacteria. Stimulation of mosquito haemocytes with *Asaia*[WSP] induced expression of cecropin, which was different from the control in *Ae. aegypti* at

three time points (6, 9, 12 h), and at only one time point in *An. stephensi* (12 h) (Supplementary Fig. 3a, b). In both *An. stephensi* and *Ae. aegypti* haemocytes we detected production of *NOS* transcripts after 9, 12 and 24 h of stimulation with *Asaia*[WSP] (Supplementary Fig. 3a, b). Finally, in both *An. stephensi* and *Ae. aegypti* none of the two bacteria determined a significant upregulation of defensin gene expression by the haemocytes, considering all the time points.

**In vivo immune gene expression**. Quantitative real-time PCR assays were used to investigate the capability of *Asaia*[WSP] to stimulate innate immune responses in *An. stephensi* and *Ae. aegypti* mosquitoes, after a sugar meal containing the engineered bacteria. To determine the dynamics of this immune response, the transcription level of immunity genes was monitored at 6, 12 and 24 h post "bacterial meal". Only female mosquitoes with fully- or partially fully-engorged abdomens were selected for these analyses. As reported in Fig. 3 (Supplementary Table 1 and Supplementary Data 1), for *Aedes* mosquitoes, four of the six analyzed genes were activated after the *Asaia*[WSP] bacteria meal (Fig. 3a). On details, cecropin D gene (*CECD*) showed an

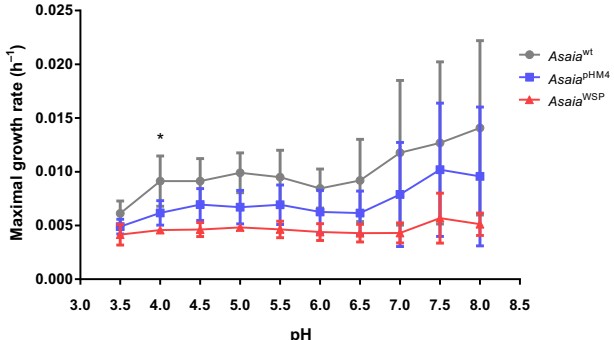

**Fig. 1 Growth rates (MGRs) of *Asaia*[wt] and recombinant strains (*Asaia*[WSP] and *Asaia*[pHM4]).** MGRs were estimated as the slope of the best regression line which fitted to the 24 h growth curves calculated for either of the strains by measuring $OD_{620}$ at ten different pH values in GLY medium. $N$ = three independent experiments were conducted. Bars represent standard deviations. Statistical analysis was carried out by Welch test with GraphPad Prism 5 software; the 24-h growth curves of wild type and both transformed strains showed differences between the mean MGRs in almost all the tested growth conditions, but these differences were not significant, with the exception of the growth at pH 4 (*$p < 0.05$, *Asaia*[WSP] vs *Asaia*[wt]).

increased expression after 12 h compared to pHM4 and sugar control; CLIP-domain serine protease gene (*CLIPB37*) resulted activated after 24 h compared to the sugar control; thio-ester containing protein 20 gene (*TEP20*) showed an upregulation on the first two time points compared to *Asaia*[pHM4] and sugar control; finally, as for NADPH-oxidases gene (*NOXM*), the gene was upregulated at all the time points, especially after 6 h. For *Aedes* mosquitoes the expression of the Transferrin gene was also investigated; after feeding with *Asaia*[WSP] a trend in the over-expression of the gene was observed, even though the differences were not significant (Fig. 3a). This agrees with results obtained on *Ae. aegypti* mosquitoes transinfected with *Wolbachia*, where the expression of this gene, involved in iron metabolism, immunity and development, is observed[15].

*Anopheles* mosquitoes that received *Asaia*[WSP] bacteria showed upregulation of *TEP1*, leucine-rich repeat protein 1 (*APL1C*), NO synthase (*NOS*) and cecropin 1 (*CEC1*)genes, compared to the controls (Fig. 3b, Supplementary Data 1). The degree and the time points of upregulation were different for the different genes: *TEP1* gene for example was upregulated after 6 and 12 h, compared to the two controls; the expression of *CEC1* gene was enhanced after 12 h; *APL1C* showed an upregulation after all the three analyzed time points, while the expression of *NOS* gene was very high after 12 h post bacterial meal (Fig. 3b). Conversely, no significant expression was detected for defensin gene in both mosquito species, in coherence with the results obtained in vitro on haemocytes.

**Mosquito colonization by engineered *Asaia*.** *Asaia* bacteria are an important and stable component of the microbiota of *An. stephensi* and *Ae. aegypti*. Here, we investigated if the transgenic bacteria were able to efficiently colonize adult *Ae. aegypti* female mosquitoes, performing an immunofluorescence assays on a total of thirty insects for each of the two different mosquito populations fed with the two engineered strains of *Asaia*. Analyses using a fluorescent confocal microscopy, after secondary staining on anti-E-tag antibodies, showed fluorescence signals inside the crop and the gut of females, indicating that the bacterium efficiently colonized these body organs. Most of the individuals showed fluorescent cells either isolated, aggregated or in microcolonies. Fluorescent cells and microcolonies were detected in the mosquito crop and gut at both 24 and 48 h after the bacterial-containing meal; colonization of the reproductive system was observed only 48 h after the meal, with very few bacteria. No immunofluorescence staining was detected in organs after the administration of *Asaia*[pHM4] strain (Figs. 4a), sugar or sugar plus kanamycin (Supplementary Fig. 4a–d and 5a, b).

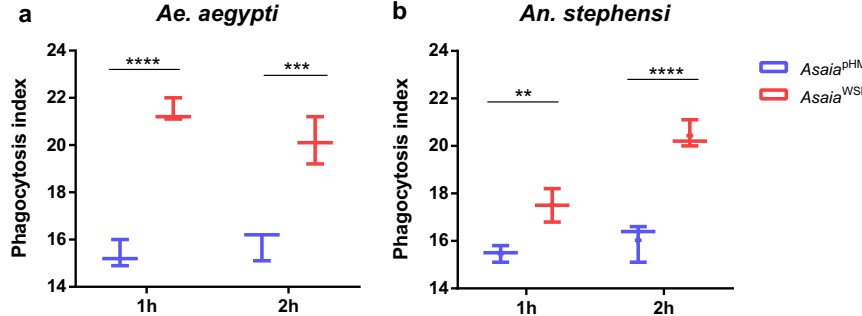

**Fig. 2 Phagocytosis tests.** Phagocytic activity was evaluated in vitro using cultured (**a**) *Ae. aegypti* and (**b**) *An. stephensi* haemocytes exposed to bacterial cells from strains *Asaia*[pHM4] and *Asaia*[WSP] incubated with FITC-fluorescent beads suspension. The percentage of haemocytes showing fluorescent phagocytised bacteria was evaluated after 1 and 2 h. Values are expressed as median±max and min of $n = 3$ replicates. $N$ = three independent experiments were conducted. Statistical significance for each experiment was determined using the two-way analysis of variance (ANOVA) followed by Sidak's multiple comparisons test where significance is represented by **$p < 0.01$, ***$p < 0.001$ and ****$p < 0.0001$.

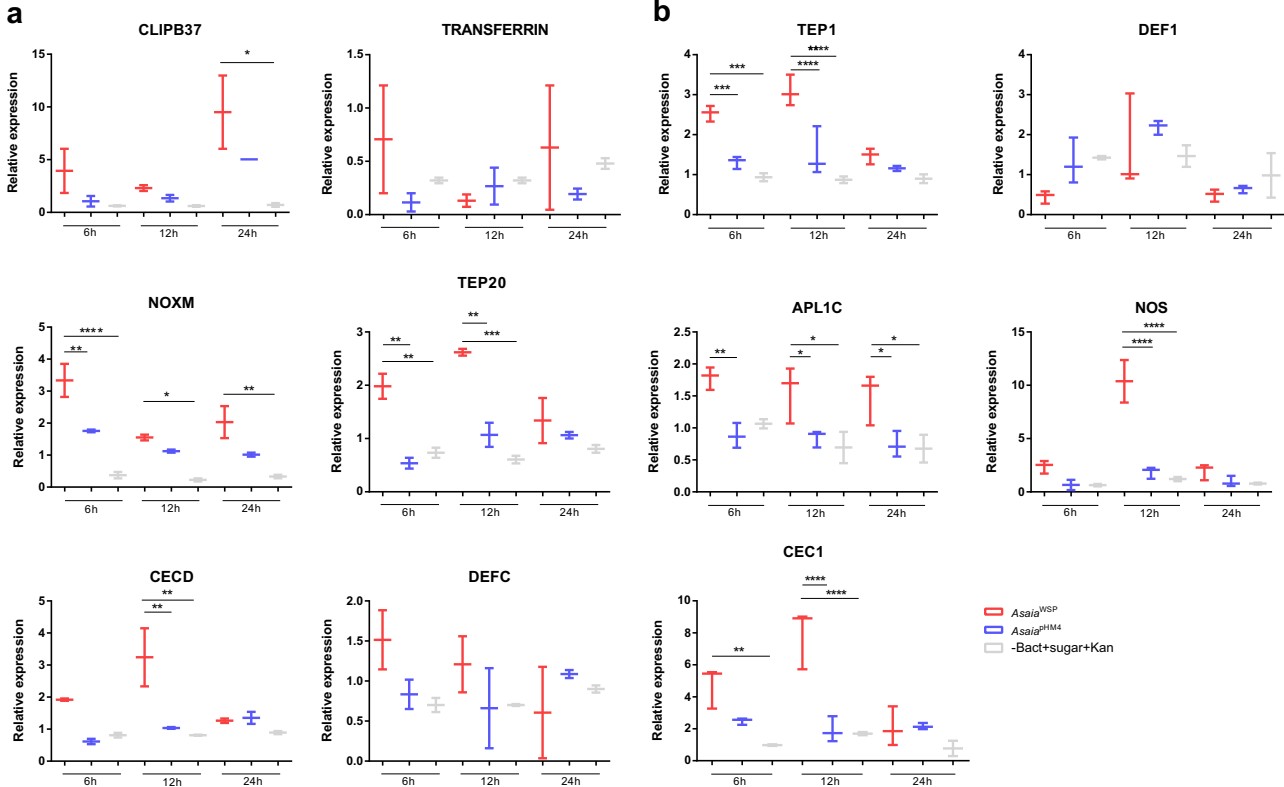

**Fig. 3 qRT-PCR analyses of differentially regulated genes. a** The differential regulation of transcript levels in *Asaia*[WSP] infected *Ae. aegypti* versus *Asaia*[pHM4] infected *Ae. aegypti* or versus mosquito fed only with sugar plus kanamycin (100 μg ml⁻¹) was examined for six selected genes: cecropin (*CECD*), transferrin (*TRANSFERRIN*), CLIP-domain serine proteases (*CLIPB37*), thioester-containing protein (*TEP20*), NADPH oxidase (*NOXM*) and defensin (*DEFC*). **b** The differential regulation of transcript levels in *Asaia*[WSP] infected *An. stephensi* versus *Asaia*[pHM4] infected *An. stephensi* or versus mosquito fed only with sugar plus kanamycin (100 μg ml⁻¹) was examined for five selected genes: cecropin (*CEC1*), *Anopheles Plasmodium*-responsive LRR protein-1C (*APL1C*), nitric oxide synthase (*NOS*), thioester-containing protein (*TEP1*) and defensin (*DEF1*). The values shown are median±max and min of at least two different qRT-PCR experiments with independent samples. Statistical analysis has been performed utilizing the analysis of variance (ANOVA) followed by Bonferroni's multiple comparisons test where significance is represented by *$p < 0.05$, **$p < 0.01$, ***$p < 0.001$ and ****$p < 0.0001$.

To quantify *Asaia* colonization, persistence and dynamics in *Ae. aegypti* (in view of the successive challenge with *D. immitis*—see below), bacteria colony-forming units of *Asaia* were assessed at different times after blood feeding, in females previously infected by the bacteria through sugar meal. In general, no statistical difference in the colonization by the two *Asaia* strains was detected (Fig. 4b). As for the pattern of organ colonization, in the midguts bacteria numbers significantly increased 24 h after the blood meal (T1, $p = 0.0068$, Fig. 4b). In the crops, the numbers decreased with time, in particular after the blood feeding (T2, $p = 0.0018$). As previously reported for *Asaia*-GFP[28], the presence of bacteria was also detected in ovaries, in coherence with the possibility of a transmission to progeny. Indeed, *Asaia*-GFP bacteria have been shown to be transmitted to progeny through and egg-smearing mechanism[4].

**Inhibition of *D. immitis* infection by *Asaia*[WSP] in *Ae. aegypti*.** Recombinant *Asaia* were administered to *Ae. aegypti* mosquitoes, Liverpool strain, through a sugar meal, 32 h before mosquitoes were fed on a *D. immitis*–infected blood meal. Three days after the blood meal we recorded an average survival rate of 35% of the mosquitoes (see Methods and Fig. 5). Figure 5 shows the results of the assay. For each group we determined two parameters: the larval abundance, i.e. the average number of L3 detected in the dissected mosquitoes; the larval prevalence, i.e. the proportion of mosquitoes that contained at least one larva at the third stage (L3), versus the total number of dissected mosquitoes. *Asaia*[WSP],

in comparison with mosquitoes fed with sugar or sugar plus kanamycin, determined a significant decrease in L3 abundance, with a reduction of 75.7% ($p < 0.0001$) and 66.8% ($p = 0.0083$), respectively. In the comparisons of mosquitoes fed on *Asaia*[WSP] with those fed on *Asaia*[pHM4], we observed reduction of 53.8%, that was however not significant ($p = 0.17$). Moreover, the feeding on *Asaia*[WSP] determined a decrease in the prevalence of L3, that was significant in the comparisons with the mosquitoes fed on sugar ($p = 0.0006$) or sugar plus kanamycin ($p = 0.0243$).

**Discussion**
Evidence has already been reported on the capability of WSP from the filarial nematode *D. immitis* to determine innate immune responses in both mosquitoes and mammals[25,27,29]. Considering the conservation of the stimuli that induce innate immunity activation across the animal phyla (e.g.[30]) and the abundance of WSP at the surface of *Wolbachia* cells[31], it is likely that this protein represents an important modulator in the interaction between the symbionts and the host in both insects and nematodes, as well as in the tripartite system *Wolbachia*-filaria-mammalian host[32]. For example, it might be a major player of the immune activation determined by *Wolbachia* in mosquitoes, as recently described[33]. Based on the above evidence and considerations, we decided to engineer a mosquito symbiont of the genus *Asaia* for the heterologous expression of WSP, in order to generate a chimeric bacterium capable of inducing

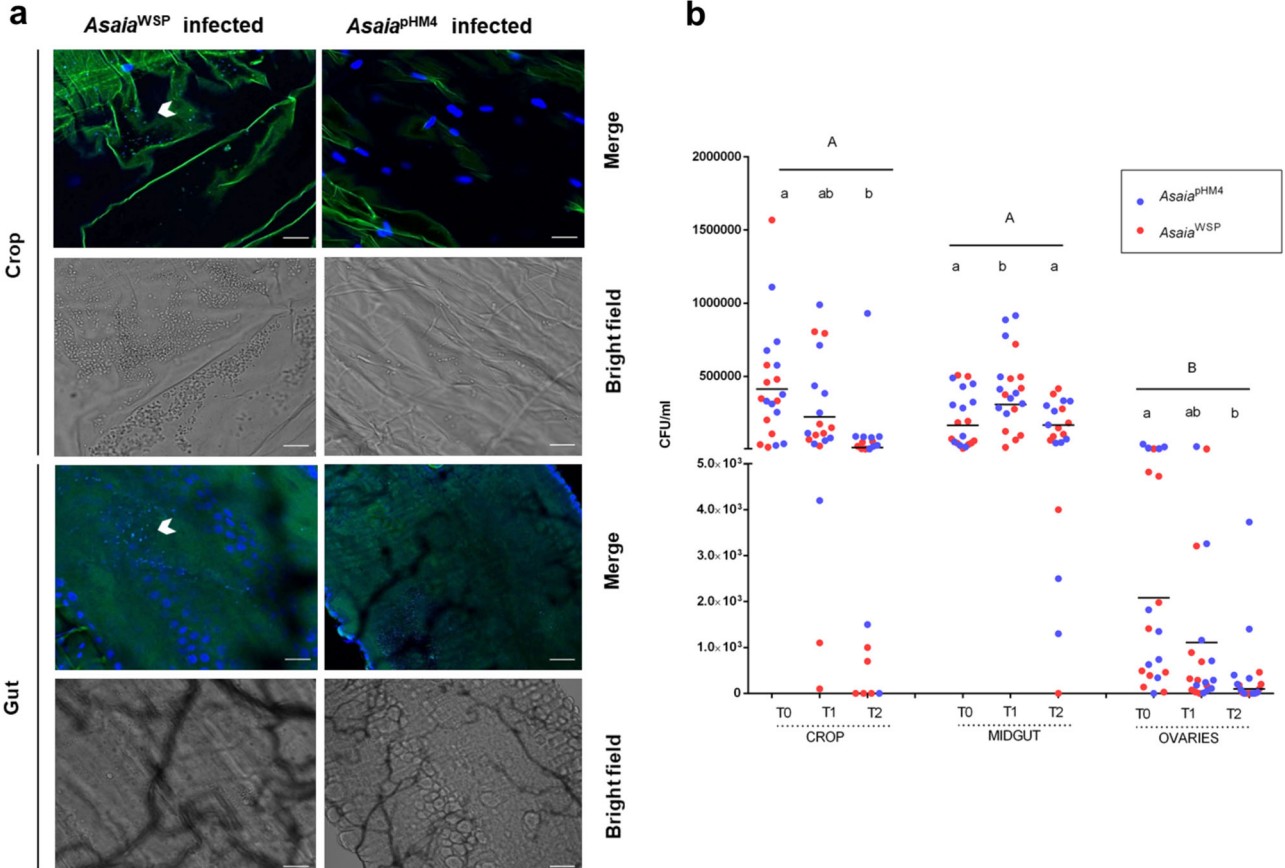

**Fig. 4 *Asaia* bacteria colonize mosquito organs and rapidly proliferate after a blood meal. a** Immunofluorescence on *Asaia*[WSP] in *Ae. aegypti* mosquitoes. *Asaia*[WSP] or *Asaia*[pHM4] were introduced to 2–3-day-old *Ae. aegypti* females via sugar meal plus kanamycin (100 μg ml$^{-1}$) for 24 h; 24 h after the bacterial meal, fed mosquitoes were selected and their organs dissected and probed with anti-E tag antibody, followed by incubation with a FITC-anti-goat IgG secondary antibody. Panels show the bright-field of the organs and the staining of *Asaia*[WSP] (white arrows indicate group of bacteria) or *Asaia*[pHM4]. Bars: 100 μm. **b** Population dynamics of *Asaia*[pHM4] and *Asaia*[WSP]. *Asaia*[pHM4] and *Asaia*[WSP] were fed to 2–3-day-old *Ae. aegypti* mosquitoes in a sugar meal for 24 h (T0), then mosquitoes were allowed to feed on a blood meal and collected after 24 (T1) and 48 h (T2). Bacteria colony-forming units were determined by plating serially diluted homogenates of organs on GLY plates containing 100 μg ml$^{-1}$ of kanamycin. The maximum bacteria number is reached when microfilariae would be invading the midgut if the blood was infected with the parasite (T1). Different capital letters represent statistically significant differences between examined organs ($p < 0.05$). Different lowercase letters represent statistically significant differences between time points in each organ ($p < 0.05$). Bars indicate the means.

immune activation in mosquito hosts and thus potentially interfering with pathogen transmission by the insect.

The first phase of the study consisted in the engineering of *Asaia* strain SF2.1 for the expression of WSP, to determine the production of the protein and to investigate the fitness of the transformed bacteria. The DNA fragment inserted into the plasmid was synthetized optimizing the codon usage and including the signal peptide, in order to allow the delivery of the recombinant protein at the surface of the bacterial cells. Western blotting and immunofluorence tests proved the expression of the protein, and the pattern of immunogold staining was coherent with a localization of the protein at the surface of the bacteria. Genetically modified microorganisms are considered as poor competitors and therefore unable to persist in the environment due to energetic inefficiency. Indeed, several studies support the idea that engineered bacteria are less fit than their native strains, but there are also examples of genetically modified organisms that display an increased fitness[34]. Therefore, the capability of *Asaia*[WSP] to grow at different pH conditions for 24 h, under continuous observation, was tested. As expected, growth rate of *Asaia*[WSP] did not surpass either those of *Asaia*[pHM4] or of the wild type strain. In fact, the growth rate of *Asaia*[WSP] was slightly slower, but the differences were not significant in all the tested conditions, but one. Thus, WSP expression does not appear to determine a significant reduction of the fitness of *Asaia*, hence a significant energetic load.

The studies published so far on the immune-modulating properties of WSP in humans, dogs, rodents and mosquitoes[25,27,29,35–37] have been conducted using a recombinant protein produced in *E. coli*, i.e. using a system that implies a possible contamination by LPS, even after highly accurate purification procedures. In the current study, immunological assays, carried out in vitro and in vivo in mosquitoes, prove to be an experimental system in which the control is very sound. The capability of *Asaia*[WSP] to induce phagocytosis and immune gene activation was higher than that of *Asaia*[pHM4], in in vitro assays. Similarly, after in vivo tests in mosquitoes, *Asaia*[WSP] increased the production of antimicrobial peptides and other immune modulators, compared to *Asaia*[pHM4] (see "Discussion" below). These results rule out the possibility that the observed higher activation of the immune response is due to contamination by LPS or other molecules, since cells and mosquitoes were stimulated with two strains of *Asaia* bacteria, with their load of LPS and other immune-modulating molecules, differing only for WSP expression. In summary, our results provide a further

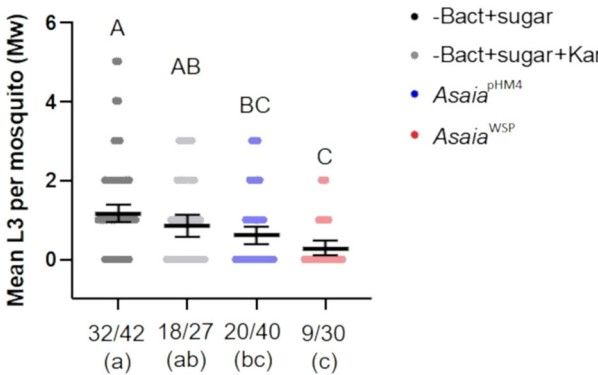

**Fig. 5 Ae. aegypti infection with transgenic bacteria and D. immitis microfilariae.** Scatter dot plots show the number of L3 larvae per mosquito. The mean numbers (±SEM) of infective L3 stage of D. immitis were determined 14 days post microfilarial challenge in Ae. aegypti by microscopical observation. Four treatments have been tested: mosquitoes fed with Asaia$^{WSP}$, Asaia$^{pHM4}$, sugar solution with or without kanamycin, before the infectious blood meal. The graph reports the average number of L3 detected in the dissected mosquitoes (abundance; y axis). The prevalence, i.e. the proportion of mosquitoes that contained at least one L3 larva versus the total number of survived and dissected mosquitoes (starting from n = 100 individuals per treatment), is shown on x axis. The geometrical mean of L3 larvae per mosquito (William's mean) was tested using a ONE-way ANOVA followed by Tukey's post-hoc test. Different capital letters, on the top of the graph, represent statistically significant differences ($p < 0.05$). The mean number of mosquitoes presenting L3 larvae was tested using contingency analysis and the final p values were adjusted using FDR. Different lowercase letters represent statistically significant differences ($p < 0.05$).

evidence of the capability of WSP to induce innate immune responses in mosquitoes.

It has been proposed that when Wolbachia is forced to create a new symbiosis with a mosquito that is naturally non infected by this bacterium, the basal immune response of the insect is enhanced, with negative effects on the mosquito's ability to transmit pathogens[33]. Mosquitoes that normally do not harbour Wolbachia, such as An. stephensi and Ae. aegypti, can therefore be regarded as good candidates to verify the immune-modulatory effects of Asaia$^{WSP}$ and, accordingly, its anti-parasite effects. In Ae. aegypti both immune deficiency- and Toll-pathways are activated by Wolbachia upon its introduction into the insect[33,38]. As for Anopheles mosquitoes, no Wolbachia had been detected in the 38 surveyed species (including An. stephensi)[39,40], till recent reports that identified Wolbachia in An. gambiae and Anopheles arabiensis[26,41]. In these local mosquito populations, naturally infected by Wolbachia, the presence of the bacterium negatively impacts Plasmodium sporozoite development[42]. In addition, studies on An. gambiae transinfected by Wolbachia suggest that Wolbachia can confer protection to mosquitoes against the pathogen Plasmodium, with an early activation of the immune response[43].

The results of our in vivo studies on An. stephensi and Ae. aegypti proved that Asaia$^{WSP}$ is able to induce an immune activation when ingested by mosquitoes. A diverse repertoire of genes coding for immune effector molecules, such as cecropin, thio-ester containing proteins, leucine-rich repeat protein and CLIP-domain serine protease, plus NADPH-oxidases and NO synthase, were upregulated in the presence of Asaia$^{WSP}$; whereas defensin levels remained unchanged. In both An. stephensi and Ae. aegypti, TEPs were among the most upregulated genes, after the bacterial meal with Asaia$^{WSP}$. We emphasize that, in

Anopheles mosquitoes, TEP induced by Plasmodium berghei binds and kills invading Plasmodium ookinetes[44]. In transgenic mosquitoes overexpressing TEP1, a reduced number of Plasmodium parasites has also been observed[45]. In Drosophila melanogaster TEPs are required for efficient phagocytosis of both Gram-positive and Gram-negative bacteria[46].

Antimicrobial peptides, that displayed upregulation in both An. stephensi and Ae. aegypti mosquitoes after feeding Asaia$^{WSP}$, are cecropins, at 6 and 12 h. Similar results (i.e. the activation of cecropin expression after stimulation with the engineered bacteria) were obtained in vitro on haemocytes from both species. It is interesting that cecropins have been shown to inhibit Plasmodium development, and also to display antiviral effects, e.g., on HIV-1[15]. In An. stephensi, the APL1C gene showed an upregulation after all the three analyzed time points; it is reported that APL1C protein is needed for protection against the rodent malaria parasites P. berghei and Plasmodium yoelii[47]. Another gene that displayed a strong upregulation (at 24 h), after the Asaia$^{WSP}$ stimulus, was CLIPB37; this result is consistent with the overexpression recorded for this gene after Wolbachia infection in Ae. aegypti and An. gambiae mosquitoes, which determined an inhibition of pathogen transmission[48,49]. The NOS gene expression has been determined both in haemocytes and in mosquitoes: our results recorded expression of NOS gene after the induction by Asaia$^{WSP}$. Mosquito NOS gene, highly homologous to the NOS genes of vertebrates, is known to be expressed during the malaria parasite invasion. In An. stephensi mosquitoes, for example, NO production has been shown to limit the development of Plasmodium parasites, in particular reducing the release of sporozoites into the hemolymph[50]. In summary, our results show that Asaia$^{WSP}$ determines an upregulation of four genes in An. stephensi, that have been recorded, in previous studies, to be involved in mosquito defence against Plasmodium spp.

A notable characteristic of Asaia bacteria is their capability to colonize mosquitoes feeding on sugar meals containing the bacteria, offering a potential tool for their introduction in the field. Our results show that Asaia$^{WSP}$ still possess the ability to colonize mosquito organs, with an increase in its abundance after the blood meal (Fig. 4). We emphasize that, in laboratory conditions, cotton pads containing Asaia bacteria, either wild type or genetically modified for WSP expression, resulted more attractive to mosquitoes than the sterile ones. Moreover, recent laboratory and field investigations, using sugar feeding as a mean for introducing bacteria different from Asaia into adult mosquitoes, highlighted that these insects are attracted by both sterile sugar solutions as well as by solutions containing the bacteria[51].

The evidence that Asaia$^{WSP}$ colonizes mosquitoes, and primes the immune response, encouraged us to test the capability of this strain to interfere with the transmission of a parasite by the mosquito themselves. As proof of principle, we focused our attention on a filarial parasite, also considering that paratransgenesis has not yet been applied in the control of insect-borne pathogenic nematodes. The model organisms, that we selected for this test, were the filarial parasite D. immitis and the mosquito vector Ae. aegypti. Our results show that Asaia$^{WSP}$ indeed interferes with D. immitis infection in mosquitoes, with differences in terms of developed L3 filarial larvae, in comparison with insects fed on sugar meals. We also recorded a difference in L3 numbers in mosquitoes fed on Asaia$^{WSP}$ in comparison with those fed on Asaia$^{pHM4}$, but this difference, although evident, was not significant. However, taken together the results on the immune priming by Asaia$^{WSP}$ in mosquitoes and the coherence in the results determined by this bacterium on filaria development in Ae. aegypti (in terms of both larval abundance and prevalence, Fig. 5), are highly encouraging. In summary, Asaia engineered for the expression of WSP can be regarded as an

inductor of innate-immune responses in mosquitoes, worth of further investigations for its potential effects on filarial parasite development.

*Asaia* bacteria have already been investigated for their capacity to interfere with pathogen transmission by mosquitoes. In a study conducted on the murine malaria model *P. berghei*, *Asaia* engineered for the expression of the scorpine antimicrobial peptide determined the inhibition of parasite infection in mosquitoes[5]. More recently, a native *Asaia* strain (SF2.1) has been shown to activate mosquito immunity, with reduction of *P. berghei* development in *An. stephensi*[52]. Our current study shows that the immune stimulating capability of *Asaia* can be boosted through the expression of a protein form *Wolbachia*. The obtained results highlight the plasticity of the *Asaia* system: the engineered bacteria, while expressing a heterologous protein at the surface, preserved their ability to colonize the insect, determining overexpression for most of the tested mosquito immune effectors. The evidence of the immune-activating capability of *Asaia*, either native[52] or genetically modified, as in the current study, requires to be validated with assays on wild-collected mosquitoes; moreover, future investigations should address the potential of *Asaia* and *Asaia*[WSP] to interfere with the transmission of arboviruses, such as dengue and Zika. Indeed, a generalized activation of mosquito immunity might imply a generalized protection of the mosquito toward infectious agents, not only filarial nematodes. Finally, safety issues should properly be addressed before proposing any strain of *Asaia* for field release. In this context, interesting investigations have been conducted on *Wolbachia* for the control of dengue virus[53], including a study on the potential transmission of this bacterium to humans[54]. We emphasize that there is strong evidence that *Asaia* colonizes the salivary glands of mosquitoes; mosquitoes might thus inoculate *Asaia* into mammals, including humans. The sole epidemiological investigation, performed so far, has not revealed any evidence for *Asaia* infection in humans, either in serological or PCR-based analyses[55]. However, we can hypothesize that *Asaia*[WSP] possesses increased immune-stimulating properties also toward humans, hence an increased pathogenic potential (e.g. proinflammatory properties[27,56]). Therefore, the issue of the potential transmission of *Asaia* to mammals would require further consideration.

## Methods

**Bacterial strains and media**. *Asaia* SF2.1 strain (*Asaia*[wt]), originally isolated from an *An. stephensi* mosquito[4], was grown at 30 °C in GLY medium (25 g L$^{-1}$ glycerol, 10 g L$^{-1}$ yeast extract, pH 5; eventually, GLY medium was solidified adding 20 g L$^{-1}$ agar). *E. coli* XL1Blue (Stratagene), used as the host for construction of plasmids, was grown at 37 °C in Luria Broth (LB; LB medium was solidified adding 15 g L$^{-1}$ agar if necessary). If needed, 100 μg/mL kanamycin was added to the media.

**Plasmid construction**. Plasmid pHM4 (≈5.5 kbp) was obtained by digesting pHM2[57] with the restriction enzyme *SacI* (Life technologies Italia). Plasmid pHM4-WSP was then constructed by inserting the WSP cassette flanked by *NotI* sites in plasmid pHM4. WSP cassette was synthesized by Eurofins Genomics (Milan) in plasmid pUC57, generating the plasmid pUC57-WSP. WSP cassette contains the neomycin phosphotranferase promoter *PnptII*, the coding DNA sequence of WSP from *Dirofilaria immitis* including the signal peptide of the gene[37], the E-TAG epitope (GAPVPYPDPLEPR[11],) and the transcription terminator *Trrn*. E-TAG epitope was inserted in the 4th loop (L4) of the *wsp* sequence[58] to allow the immunodetection of the expressed protein. Moreover, the *wsp* gene sequence was optimized according to the codon usage of strain SF2.1 as inferred from its genome sequence[59]. WSP cassette was then digested from pUC57-WSP by using the restriction enzyme *NotI*, loaded in 1% agarose gel and purified by using QIAquick Gel Extraction Kit (Qiagen). Plasmid pHM4 was digested with *NotI*, dephosphorylated by using Shrimp Alkaline Phosphatase (SAP, Life technologies Italia) and the *wsp* fragment was ligated to the *NotI*-linearized pHM4 by using T4 DNA ligase (Life technologies Italia). Ligation product was then used to transform *E. coli* XL1Blue electrocompetent cells[4]. Recovery was performed with LB medium for 1 h at 37 °C with shaking before plating on LB plates added with kanamycin. Putative transformants were selected and successful ligation of WSP cassette was checked by PCR using *wsp*-specific primers (see Supplementary Table 1). The obtained plasmid, named pHM4-WSP (Supplementary Fig. 1a), was then extracted

from *E. coli* and electroporated in *Asaia*[wt] as previously described[4] resulting in the strain *Asaia*[WSP]. Strain *Asaia*[pHM4] was also obtained and used as control in the following experiments.

**Western-blot detection of WSP produced by *Asaia* strains**. For protein secretion, the *Asaia*[WSP] and *Asaia*[pHM4] were grown overnight at 30 °C in GLY medium supplemented with 100 μg ml$^{-1}$ of kanamycin. Bacterial cultures were centrifuged at $3000 \times g$ for 15 min at RT and pellets were resuspended in SDS sample loading buffer 1×; a protease inhibitor cocktail was also added to avoid the protein degradation. Briefly, membranes were blocked in blocking buffer (4% milk in PBS with 0.1% Tween 20) and probed with the primary goat anti-E tag antibody (Novus Biologicals), followed by an HRP-conjugated anti-goat IgG secondary antibody.

**RNA extraction and reverse transcription-quantitative PCR**. Bacteria were grown at OD 0.5, 1, 1.5 (three pools each) and stored in RNAprotect Bacteria Reagent (Qiagen); RNAs were extracted using RNeasy Mini Kit (Qiagen) including an on-column DNase I treatment to remove residual DNA. RNA was stored at −80 °C till further use. RNA purity was checked by determining the 260/280 nm absorbance ratio. cDNAs were synthesized from 250 ng of total RNA using a QuantiTect Reverse Transcription Kit (Qiagen) with random hexamers. The cDNA was used as template in RT-PCR reactions. Quantitative RT-PCRs on *Asaia*[WSP] and *Asaia*[pHM4] were performed under the following conditions: 100 ng cDNA; 250 nM of forward and reverse primers (target gene *wsp*; see Supplementary Table 1 for primers sequences); 98 °C for 30 s, 40 cycles of 98 °C for 15 s, 58 °C for 30 s; fluorescence acquisition at the end of each cycle; melting curve analysis after the last cycle. The quantification cycle values were determined, in order to calculate gene expression levels of the target gene relative to 16S rRNA, the internal reference gene for *Asaia*[55]. The estimates of the expression level of *wsp* gene has been reported as the means ± standard error (SEM).

**In vitro growth assays of *Asaia* for fitness measurements**. Growth assays of *Asaia* strains were performed at different pH values (from 3.5 to 8.0 with increases of 0.5). Bacterial cells were grown overnight at 30 °C with constant agitation (130 rpm) in GLY broth. For recombinant strains, 100 μg ml$^{-1}$ kanamycin was added to the medium. For each strain, a dilution to 0.1 optical density at 620 nm (OD$_{620}$) was carried out in GLY medium at different pH (range 3.5–8.0) and 200 μl were distributed in 96-well microtiter plates wells (two wells for each condition). Growth was recorded by an EnSight plate reader (Perkin Elmer), measuring the OD$_{620}$ in each well every 10 min for 24 h at 30 °C. As negative control, growth medium without bacteria was used. OD$_{620}$ values were collected and, after baseline correction, the maximal growth rate (MGR) (h$^{-1}$) was estimated as the slope of the best regression line which fitted to growth curve, for either of the strains during the time interval. Growth assays were repeated three times. MGRs were compared by strain and medium pH using a Welch test. Student's *t*-test (two-sides, Welch's correction) was performed by GraphPad Prism 5 software. $P < 0.05$ was considered significant.

**Immunofluorescence assays on bacteria and on mosquitoes**. Recombinant *Asaia* expressing WSP or with plasmid alone were grown as reported above; 10 μL of a cell suspension at the concentration of $10^8$ cells ml$^{-1}$ in PBS were placed on glass slides, air dried, and fixed for 20 min with cold methanol. Bacterial cells were blocked in bovine serum albumin (FBS) and probed with the primary goat anti-E tag antibody (Novus Biologicals), followed by incubation with an anti-goat IgG secondary antibody, FITC Conjugate (Sigma-Aldrich).

As for the detection of transgenic bacteria in mosquito organs, *Asaia*[WSP] or *Asaia*[pHM4] were administered to 2–3-day-old *Ae. aegypti* (Liverpool black-eyed strain) females via sugar meal ($1 \times 10^8$ cells ml$^{-1}$) plus kanamycin (100 μg ml$^{-1}$) for 24 h; 24 h after the bacterial meal, fed mosquitoes were selected and their organs (crop, midgut and ovaries) dissected and fixed in 4% (wt vol$^{-1}$) paraformaldehyde at 4 °C, washed in PBS, and blocked with 4% (wt vol$^{-1}$) FBS. The samples were then probed with goat anti-E tag antibody, followed by incubation with a FITC-anti-goat IgG secondary antibody. Observations were recorded with a Leica microscope (LeicaTCSNT) and analyzed with ImageJ software. Survival of mosquitoes was also monitored daily. Survival percentages represent the mean survival percentage of three biological replicates of 30 mosquitoes each.

**Immunogold staining on bacteria pure culture**. *Asaia*[WSP] or *Asaia*[pHM4] samples were fixed by immersion in 4% paraformaldehyde in PBS for 2 h at 4 °C and washed in PBS. Free aldehydes were blocked in 0.5 MNHCl in PBS for 45 min at 4 °C; samples were washed in PBS, dehydrated through graded concentrations of ethanol and embedded in LR White resin (Electron Microscopy Sciences) overnight, at 4 °C. Resin samples were polymerized for 24 h at 60 °C. Ultrathin sections were placed on grids coated with a Formvar-carbon layer and then processed for immunocytochemistry. Ultrathin sections were floated for 3 min on normal goat serum (NGS) diluted 1:100 in PBS and then incubated overnight at 4 °C with goat anti-E tag antibody diluted with PBS containing 0.1% BSA and 0.05% Tween 20. After rinsing, sections were floated on NGS and then reacted for 20 min at room temperature with secondary 12 nm gold-conjugated antibodies (Jackson Laboratories) diluted 1:20 in

PBS. The specimens were observed on a Philips Morgagni transmission electron microscope operating at 80 kV and equipped with a Megaview II camera for digital image acquisition.

**Colonization and quantification of *Asaia* in mosquitoes.** To investigate colonization of *Asaia* in different tissues of mosquitoes, 2–3-day-old adult mosquitoes were fed for 24 h on a cotton pad moistened with 5% sterile sucrose solution containing $10^8$ cells ml$^{-1}$ bacteria (T0). The bacteria-fed mosquitoes were starved for 10 h, and then allowed to feed on a blood meal. Twenty-four (T1) and 48 h (T2) after the blood meal, the individual mosquitoes were surface-sterilized by washing them in 75% ethanol for 3 min and then rinsing them in sterile PBS three times. The crop, midgut and ovaries were dissected under sterile conditions and homogenized in 0.2 ml sterile PBS. The bacterial load was determined by plating tenfold serial dilutions of the homogenates on GLY plates containing 100 μg ml$^{-1}$ of kanamycin and incubating the plates at 30 °C for 48 h. The colonies were counted and the data analyzed using RStudio. Briefly, a three-way ANOVA was used to test the global variance of the data and to assess which of the three categorical independent variables (*Asaia* strain, time and mosquito tissues) influences the *Asaia* load. After having assessed that the *Asaia* strain did not affect the colonization of the mosquito tissues, we performed a two-way ANOVA (using time and mosquito tissues as categorical independent variables) to test the interactions of the variables and one-way ANOVA to analyse the variance of *Asaia* within each of the tissues over time.

**Haemocyte primary cultures and phagocytosis test.** Mosquito haemocytes were isolated from dissected *An. stephensi* and *Ae. aegypti* adults and maintained 72 h in Schneider's medium (Sigma-Aldrich), supplemented with heat-inactivated 10% fetal bovine serum (FBS), 100 units ml$^{-1}$ penicillin and 100 μg ml$^{-1}$ streptomycin, before further analyses. Antibiotics have been removed before the phagocytosis test by centrifugation of cells and resuspending them in fresh medium without any addition. In the phagocytic tests, haemocyte cultures from both mosquito species were incubated for 6 h in 1 ml of medium containing bacteria. Successively, haemocytes were shortly centrifuged, resuspended in 200 μl of fresh Schneider's medium (without any supplement) and then incubated with 0.1 μl of a FITC-fluorescent beads suspension for 1 and 2 h in soft oscillation, according to ref. [60]. After incubation, cells were cytocentrifugated onto glass slides, counterstained with a 200 ng ml$^{-1}$ propidium iodide solution and observed with a Zeiss Axioplan epifluorescence microscope. The phagocytosis index was evaluated as the percentage of haemocytes showing inside fluorescent particles. Three phagocytic test replicated experiments were performed. Statistical analysis has been performed using GraphPad Prism 5 utilizing the two-way analysis of variance (ANOVA) followed by Sidak's multiple comparisons test ($p < 0.05$ has been considered significant).

**Antimicrobial peptides and nitric oxide synthase expression in hemocytes.** *An. stephensi* and *Ae. aegypti* hemocytes has been incubated with a $10^9$ cells ml$^{-1}$ bacterial solution for 0, 3, 6, 9, 12 and 24 h. After treatments, cells were centrifuged at $800 \times g$ for 5 min at room temperature and the supernatant was discarded. Total RNA was extracted from cells using TRI-REAGENT TM (Sigma), following the method described by the supplier. RT-PCR has been performed with the Access RT-PCR System (Promega), according to the supplier's protocols. For *An. stephensi* and *Ae. aegypti, actin* was used as reference gene; the sequences of the analyzed genes and the relative citations were reported in Supplementary Table 1. For both species, PCR amplification gel documentation was collected using a Gel Doc XR, digitally evaluated with Quantity One (Bio-Rad Lab) and normalized to the correspondent signals for cytoplasmic actin. Three replicates were carried out for each induction.

**Immune gene expression in mosquitoes fed with bacteria.** *Asaia*$^{WSP}$ or *Asaia*$^{pHM4}$ were administered to 2–3-day-old adult female mosquitoes (*An. stephensi* and *An. aegypti*) via sugar meals, bred in small cages containing 50 samples. Mosquitoes were allowed to feed for 6, 12 and 24 h on a sterile cotton pad moistened with 5% sterile sucrose solution containing $10^8$ cells ml$^{-1}$ bacteria (plus kanamycin 100 μg ml$^{-1}$), or 5% sugar plus kanamycin with no bacteria (as control).

After 6, 12 and 24 h, the mosquitoes were collected and stored in RNA later at −80 °C for RNA extraction and molecular analysis. The expression profiles of 11 immune-related genes (see Supplementary Table 1 for primer sequences and details), were analyzed by quantitative RT-PCRs. Briefly, RNA was extracted from pool of three mosquitoes using the RNeasy Mini Kit (Qiagen), according to the manufacturer's instructions. cDNAs were synthesized from 150 ng of total RNA using a QuantiTect Reverse Transcription Kit (Qiagen). Quantitative RT-PCRs on target genes were performed using a BioRad Real-Time PCR Detection System (Bio-Rad) at the following conditions: 50 ng cDNA; 300 nM of forward and reverse primers; 98 °C for 30 s, 40 cycles of 98 °C for 15 s, 56-60 °C for 30 s; fluorescence acquisition at the end of each cycle; melting curve analysis after the last cycle. In order to calculate the expression of the target genes, quantification cycle (Cq) values were determined for each gene and normalized according to the endogenous reference genes *rps7* or *rps17* (Supplementary Table 1). The estimates of the

expression level of each gene are relative to the control groups and reported as fold change mean ± standard error mean (SEM) of at least three replicates. Statistical analysis has been performed using GraphPad Prism 5 utilizing the ANOVA followed by Bonferroni's multiple comparisons test ($p < 0.05$ has been considered significant).

**Mosquito infection with bacteria and microfilariae.** Microfilariaemic blood samples from a dog naturally infected with *D. immitis* and blood from an uninfected dog were kindly provided by Prof. Genchi; bloods were anticoagulated with heparin. Since blood was collected for diagnostic purposes and the owners signed an informed consent that authorize the use of residual samples (i.e. the amount of blood remained after diagnostic clinical chemistry) for research purposes, according to the regulations of our Institution (EC decision 02-2016) a formal approval from the Ethical Committee was not required. Vitality and number of *D. immitis* microfilariae in all samples were confirmed by microscopy; briefly, 20 μl of blood were mixed with 40 μl of distilled water, covered with a cover slide, and microfilariae were counted by examination with a microscope (4×). Microfilaraemiae of the dog was determined three times. For the inoculation experiments, *Ae. aegypti* female mosquitoes at an age of 2–3 days were selected, maintained at standard condition in cages of 100 samples[61] and fed on a sterile cotton pad moistened with 5% sucrose solution containing $10^8$ cells ml$^{-1}$ bacteria (with kanamycin 100 μg ml$^{-1}$) for 1 day (four treatments: mosquitoes fed with *Asaia*$^{WSP}$ or *Asaia*$^{pHM4}$ plus microfilariae, sugar solution with/without kanamycin plus microfilariae).

Microfilariaemic counts were adjusted to 3500 mf ml$^{-1}$ with blood from uninfected dog. The microfilaria load in the infecting blood was according to recommended protocols[62], in order to avoid an excess in larval mortality, caused by nematode larvae. Sugar was removed and the mosquitoes were allowed to feed through Parafilm® membranes for at least 1.5 h on 5 ml blood at 37 °C in an artificial feeding system. Three to five mosquitoes were immediately dissected to verify mean microfilariae ingested per mosquito. Mosquitoes were kept for up to 14 days in cages with access to 5% glucose and water ad libitum; after this time, mosquitoes were collected and exposed for 2 min in a freezer for immobilization, and the wings and legs were removed. Only the mosquitoes for which a blood meal was completed were collected. These mosquitoes were dissected individually: the abdomen was separated and midgut contents were smeared on a slide; *D. immitis* L3 larvae were thus counted. Statistical analysis was performed by GraphPad Prism 5 software. The mean number of individuals presenting L3 larvae was tested using contingency analysis and the final p values were adjusted using FDR, while the geometric mean number of L3 larvae per infected mosquito was calculated using the William's mean (Mw)[63], considering the high proportion of mosquitoes not presenting L3 larvae, and Mw were analyzed using a ONE-way ANOVA followed by Tukey's post-hoc test.

**Reporting summary.** Further information on research design is available in the Nature Research Reporting Summary linked to this article.

## Data availability
The datasets generated during and/or analyzed during the current study are available from the corresponding author (Molecular and Evolutionary Parasitology Lab, Department of Biosciences, University of Milan), on reasonable request. The source data underlying plots shown in figures are presented in Supplementary Data 1.

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

## Acknowledgements

This study was supported by Cariplo Foundation and Lombardy Region to S.E. (2017-N.1656), by Transition Grant 2015-2017-Horizon 2020 to S.E. (G42F17000140001) and by the MIUR PRIN (Italian Ministry of Education, University and Research, MIUR) to C.B., L.P. and G.F. D.D. acknowledges the baseline funding support by King Abdullah University of Science and Technology (KAUST); S.E., C.B. and M.B. thank "Fondo per il finanziamento dei Dipartimenti universitari di eccellenza" (MIUR) of the Dept. of Biosciences (University of Milan) and of the Dept. of Biology and Biotechnology "L. Spallanzani" (University of Pavia). The authors thank Prof. L. Sacchi, Prof. F. Forlani and

Prof. C. Bazzocchi for their suggestions and the UNITECH platform for the microscope image acquisition.

## Author contributions

S.E. and I.V.B. performed and supervised the majority of experiments. S.E. and E.C. generated transgenic bacteria; I.V.B., L.G. and L.P. conducted fitness experiments; M.B. performed immunogold assays and M.M. conducted the in vitro experiments; C.D., P.G. and M.G. performed the cage experiments and transmission-blocking assays; G.F., D.D. and C.B. analyzed the data; S.E., I.V.B. and C.B. wrote the manuscript; all authors revised and provided inputs to the manuscript.

## Competing Interests

The authors declare no competing interests.
