## [Peer Review File · Communications Biology]

Reviewers' comments:

Reviewer #1 (Remarks to the Author):

1- There is no doubt that the presence of Wolbachia symbionts (or WSP expressed by Asaia bacteria) will have an effect on mosquito physiology. However, the main question is what effect, and more importantly, is that the desired effect?

See the paper below as an example of Wolbachia infection enhancing West Nile virus Infection rather than reducing it.

Dodson BL, Hughes GL, Paul O, Matarachiero AC, Kramer LD, Rasgon JL. Wolbachia Enhances West Nile Virus (WNV) Infection in the Mosquito *Culex tarsalis*. *PLoS Negl Trop Dis*. 2014;8: e2965.

doi:10.1371/journal.pntd.0002965

2- There are no issues with the experiments and expression/production of WSP by Asaia bacteria. However, I have concerns about the authors' choice in testing the inhibition of *D. immitis* infection by AsaiaWSP in *Ae. aegypti*?

Why choosing *D. immitis* with virtually no epidemiological relevance over Zika, dengue, and chikungunya? How can the authors conclude that "The modified Asaia, here presented, appears as a promising tool in which, in addition to the peculiarities and advantages offered by Asaia, immune activation determined by WSP could be effective to reduce the transmission of a variety of vector-borne pathogens, not only filariae, from various insects" by testing the effect of AsaiaWSP on *D. immitis*?

Moreover, the conclusions are too broad and not supported by the result. "Our results provide a further evidence for the capability of WSP to induce innate immune responses in insects." This will have to be addressed.

3- "The degree and the time points of upregulation were different for the different genes: 173 TEP1 gene for example was significantly upregulated after 6 and 12 hours, compared to the two controls; the expression of CEC1 gene was enhanced after 12 hours; APL1C showed a statistically significant upregulation after all the three analysed time points, while the expression of NOS gene was very high after 12 hours post bacterial meal." What does that mean from Plasmodium development? What would be the effect of these genes in Plasmodium development in *An. stephensis*? Will it have different effects accordingly to different Plasmodium species?

4- The results obligatorily will have to be validated in field-collected mosquitoes. Lab strains such as the Liverpool strain constantly suffer from genetic drift events and are not representative of wild-collected mosquitoes, which at the end are the target of any control strategy. Lab mosquitoes may be used as a model but will not provide definitive results. Furthermore, using *D. immitis* will impair any conclusion on this particular technique on arbovirus, and again, will not provide definitive results.

5- The discussion is broad and there are major overlaps with the introduction such as in lines 258-275. It is also a bit confusing. It would be nice to follow the same logic of the results by discussing: (i) Expression/production of WSP by Asaia SF2.1 and fitness of the engineered bacteria; (b) In vitro phagocytosis test and immune-related gene expression; (c) In vivo immune gene expression; and (d) Mosquito colonization by engineered Asaia.

6- The use of genetically modified bacteria to control pathogens is years if not decades away from being used on the field. See VCAG (<https://www.who.int/vector-control/vcag/en/>). Please address this issue in the discussion and conclusion. See the example below.

"The use of genetically modified bacteria to control pathogens (expressing anti-pathogen molecules or molecules able to activate the immune system) can be considered a strategy compatible with current

mosquito control tools (insecticides, SIT, Wolbachia replacement) or with genetically modified mosquitoes.”

Reviewer #2 (Remarks to the Author):

In search for a more effective vector-harnessed biocontrol of mosquito-borne diseases, the study investigated the anti-pathogen effect of Wolbachia surface protein (WSP) expressed by an engineered bacterial endosymbiont *Asaia* in mosquitoes. The general idea is interesting and the study will make valuable contribution to the scientific community. My observations and suggestions are below.

1. Line 73: "Indeed, trough..." Through?
2. Line 78: "Aedes aegypti-mosquitoes..." remove hyphen
3. It will be good to provide suitable reference(s) to the statements in line 81-83.
4. Line 91-92: "...supporting the activity of this protein as a general trigger of innate immune activation both in insects and in mammals." This sentence, at it is, seems to be left open to doubt about human safety. One may wonder that, since WSP which is said to be capable of inducing innate immune response in mammals (including humans) may also be present in mosquito saliva and injected into humans during mosquito bloodfeeding (see reference below), could being bitten by this type of engineered cause increased allergic reactions in humans? What is the authors' view on this? How can the authors clarify this concern? The authors might want to see a previous discussion by Popovici et al 2011 (<http://dx.doi.org/10.1590/S0074-02762010000800002>). In this kind of paper, the genetic manipulation technique of focus becomes more fascinating if the text reflects some potential safety concerns that have been experimentally eliminated.

Reviewer #3 (Remarks to the Author):

Based on the contribution of immune priming to Wolbachia-mediated pathogen interference, the authors expressed Wolbachia surface protein (WSP), a potential inducer of immunity response, in *Asaia* bacteria, and observed activation of a number of immune antiviral or anti-plasmodia immune genes after introducing the recombinant bacteria (*Asaia*WSP) into mosquito. Furthermore, the development of the heartworm parasite *Dirofilaria immitis* was inhibited by *Asaia*WSP in *Ae. aegypti*. Overall, this is great work with experimental designed logically. It can facilitate not only better understanding of the mechanism of Wolbachia-mediated pathogen interference in mosquito, but also develop a new artificial symbiont for vector disease control. Thus, I support its publication by *Communications Biology* after addressing the below comments. In addition, the writing of this manuscript should be largely improved to make it readable before publishing.

Fig. S1, e and f should be list together with Fig. S2, a and b in the same panel for easy comparison. The statistic information for Fig. S3 should be indicated.

Line 166 and 167, if not statistical significantly, Transferrin should not be thought as over-expressed. It may be ok to say a trend in over-expression was observed.

Fig. S4, legend should indicate which figures are 24 hs and 48 hr. Fig 4b, it is better to present the same tissue in parallel for comparison. It is not easy to see the positive signal in Crop and Gut for *Asaia*WSP treatment as it is presented currently. The results of Fig 4a should be mentioned in the text

before Fig 4b, or the figure label should be switched. In Fig 4a, the medium value should be indicated using a horizontal line.

Fig 5, what is the goal to include additional control using sugar and kanamycin?

214-224, please conclude a reduction only when there is statistic difference. How to interpret no difference in both abundance and prevalence between AsaiaWSP, AsaiapHM4 treatment?

73, trough should be through

77, ";" should be behind citation.

82, "In addition, it is likely not very resistant outside cells". It is difficult to understand this sentence

100-102, these sentence should belong to discussion

Point-by-point replies to the reviewers' comments (COMMSBIO-19-1406-T)

Please note: in the manuscript, sentences that have been removed are in triple square brackets and highlighted in red; new sentences, or thoroughly revised parts of the manuscript, are highlighted in yellow. The flow of the arguments presented in the discussion has been reorganized (as suggested by Reviewer 1); the whole discussion is thus highlighted in yellow. Moreover, the reference list has been adjusted based on the introductions (or removal) of a few citations, and on the reorganization of paragraphs in the manuscript.

Reviewer #1 (Remarks to the Author):

1- There is no doubt that the presence of *Wolbachia* symbionts (or WSP expressed by *Asaia* bacteria) will have an effect on mosquito physiology. However, the main question is what effect, and more importantly, is that the desired effect? See the paper below as an example of *Wolbachia* infection enhancing West Nile virus Infection rather than reducing it.

Dodson BL, Hughes GL, Paul O, Maticchiero AC, Kramer LD, Rasgon JL. *Wolbachia* Enhances West Nile Virus (WNV) Infection in the Mosquito *Culex tarsalis*. *PLoS Negl Trop Dis*. 2014;8: e2965. doi:10.1371/journal.pntd.0002965

We agree that the effects of *Wolbachia* infection on the host physiology are not predictable. We thus introduced a sentence to address this issue in the Introduction section (highlighted in yellow) and quoted the paper by Dodson and colleagues (2014).

2- There are no issues with the experiments and expression/production of WSP by *Asaia* bacteria. However, I have concerns about the authors' choice in testing the inhibition of *D. immitis* infection by *Asaia*WSP in *Ae. aegypti*?

Why choosing *D. immitis* with virtually no epidemiological relevance over Zika, dengue, and chikungunya? How can the authors conclude that "The modified *Asaia*, here presented, appears as a promising tool in which, in addition to the peculiarities and advantages offered by *Asaia*, immune activation determined by WSP could be effective to reduce the transmission of a variety of vector-borne pathogens, not only filariae, from various insects" by testing the effect of *Asaia*WSP on *D. immitis*? Moreover, the conclusions are too broad and not supported by the result. "Our results provide a further evidence for the capability of WSP to induce innate immune responses in insects." This will have to be addressed.

The assays on *Ae. aegypti*, on the inhibition of *D. immitis* infection by *Asaia*WSP, were carried out in order to reply to the question: is the immune activation that we observed (in terms of enhanced phagocytosis and expression of antimicrobial peptides and NOS) effective toward a pathogen? Therefore, we did not focus on a highly epidemiologically important pathogen; rather we followed the seminal paper by Kambris et al. 2009 (Science), which indicated that immune activation by *Wolbachia* determines the inhibition of filarial L3 development. In other words, we considered filarial nematodes as a suitable model to investigate the effects of mosquito immune activation. In terms of the epidemiological importance, *D. immitis* is still an important parasite in dogs, also in areas

where *Ae. aegypti* acts as a vector. We agree with Reviewer 1 that future studies should focus on viruses, and we emphasized this point at the end of the discussion.

We agree with Reviewer 1 that the conclusions are too broad: while we believe that the activation of mosquito immunity that we recorded, in the two mosquito models, is interesting beyond the specific effects on *D. immitis*, we are convinced that we cannot derive conclusions about the possible effects on other pathogens. We have thus thoroughly revised the discussion of the manuscript, in the light of the reviewer's comment.

3- “The degree and the time points of upregulation were different for the different genes: 173 TEP1 gene for example was significantly upregulated after 6 and 12 hours, compared to the two controls; the expression of CEC1 gene was enhanced after 12 hours; APL1C showed a statistically significant upregulation after all the three analysed time points, while the expression of NOS gene was very high after 12 hours post bacterial meal.” What does that mean from Plasmodium development? What would be the effect of these genes in Plasmodium development in *An. stephensis*? Will it have different effects accordingly to different Plasmodium species?

We have not yet planned an experiment on the inhibition of *Plasmodium* development by *Asaia*WSP. In the experiments on *D. immitis* we cannot determine the exact hour at which mosquitoes ingested the sugar meal containing the bacteria (the cotton pad was available to mosquitoes for 24 hours). However, since *Asaia* load significantly increases after the blood meal (Fig. 4), we could assume that, in a mosquito already colonized by *Asaia*WSP, these bacteria could determine a stimulation of immunity following the blood meal. We could thus suggest that, in parallel with *Plasmodium* infection in the mosquito, *Asaia* would possibly bloom, stimulating the immunity. Considering that the peak in the *Asaia* load is reached at 24 hours after the blood meal (as observed in the current study), the maximum load in *Asaia* approximately corresponds to the time of ookinete invasion of the gut wall. The phase of ookinete invasion of the gut, and of its successive localization toward the haemocoel is likely suitable for an attack by the immune system, and there is indeed evidence that TEP interferes with ookinete invasion in *Plasmodium berghei*. In addition, there is evidence that NO limits the release of sporozoites from the oocyst. Made these considerations, which suggest that *Asaia*WSP could potentially interfere with *Plasmodium* transmission, it would be rather difficult to discuss this issue in the manuscript, in the absence of precise picture on the timing of *Asaia* ingestion by mosquitoes in relation with the time of *Plasmodium* infection. We however agree with reviewer 1 that the potential effect of *Asaia*WSP on *Plasmodium* would deserve some further comments. We thus added a couple of sentences to discuss the upregulation of genes that could be involved in mosquito defense against *Plasmodium*.

4- The results obligatorily will have to be validated in field-collected mosquitoes. Lab strains such as the Liverpool strain constantly suffer from genetic drift events and are not representative of wild-collected mosquitoes, which at the end are the target of any control strategy. Lab mosquitoes may be used as a model but will not provide definitive results. Furthermore, using *D. immitis* will impair any conclusion on this particular technique on arbovirus, and again, will not provide definitive

results.

As already discussed at point 1, the goals of this work were to: i) provide further evidence on the immune-stimulating capacity of WSP; ii) determine whether a symbiont manipulated for the expression of this molecule could stimulate immunity in mosquitoes; iii) verify whether, in parallel with the immune activation, there is also some impairment on the development of a parasite. In other words, we did not intend to provide a tool suitable to be used in paratransgenesis, rather our goal was to obtain a first evidence on the capability of a bacterial vehicle, engineered for the expression of *Wolbachia* molecule, to stimulate mosquito immunity.

We therefore agree with reviewer 1: our results are far from providing definitive results in terms of possible applications. Further experiments on wild-collected mosquitoes and in semi-field conditions will be required, before the strategy could be discussed for potential applications. We thus revised the discussion, and part of the introduction, in order to avoid giving too much emphasis to the potential applications of our results in paratransgenesis, and to highlight that results obtained on lab strains need to be validated on wild-collected mosquitoes (see also reply at point 1).

5- The discussion is broad and there are major overlaps with the introduction such as in lines 258-275. It is also a bit confusing. It would be nice to follow the same logic of the results by discussing: (i) Expression/production of WSP by Asaia SF2.1 and fitness of the engineered bacteria; (b) In vitro phagocytosis test and immune-related gene expression; (c) In vivo immune gene expression; and (d) Mosquito colonization by engineered Asaia.

Done as suggested. We thoroughly revised the discussion in order to avoid overlaps with the introduction. In addition, we reorganized the discussion, in order to be coherent with the presentation of the results. The current version of the discussion contains: i) an opening paragraph; ii) the discussions of the results, in the same order of the results section; iii) a concluding paragraph.

6- The use of genetically modified bacteria to control pathogens is years if not decades away from being used on the field. See VCAG (<https://www.who.int/vector-control/vcag/en/>). Please address this issue in the discussion and conclusion. See the example below.

“The use of genetically modified bacteria to control pathogens (expressing anti-pathogen molecules or molecules able to activate the immune system) can be considered a strategy compatible with current mosquito control tools (insecticides, SIT, *Wolbachia* replacement) or with genetically modified mosquitoes.”

The discussion has been thoroughly revised, following the suggestions of the reviewers. The emphasis on paratransgenesis was deeply reduced, and the above sentence was removed from the discussion. Also, the issue of the applicability of GM bacteria has been addressed.

Reviewer #2 (Remarks to the Author):

In search for a more effective vector-harnessed biocontrol of mosquito-borne diseases, the study investigated the anti-pathogen effect of Wolbachia surface protein (WSP) expressed by an engineered bacterial endosymbiont *Asaia* in mosquitoes. The general idea is interesting and the study will make valuable contribution to the scientific community. My observations and suggestions are below.

1. Line 73: "Indeed, trough..." Through?

Corrected

2. Line 78: "Aedes aegypti-mosquitoes..." remove hyphen

Removed

3. It will be good to provide suitable reference(s) to the statements in line 81-83.

A reference is now provided (Rasgon et al, 2006 Appl Environ Microbiol).

4. Line 91-92: "...supporting the activity of this protein as a general trigger of innate immune activation both in insects and in mammals." This sentence, at it is, seems to be left open to doubt about human safety. One may wonder that, since WSP which is said to be capable of inducing innate immune response in mammals (including humans) may also be present in mosquito saliva and injected into humans during mosquito bloodfeeding (see reference below), could being bitten by this type of engineered cause increased allergic reactions in humans? What is the authors' view on this? How can the authors clarify this concern? The authors might want to see a previous discussion by Popovici et al 2011

(<http://dx.doi.org/10.1590/S0074-02762010000800002>). In this kind of paper, the genetic manipulation technique of focus becomes more fascinating if the text reflects some potential safety concerns that have been experimentally eliminated. We thank the reviewer for suggesting to discuss the safety issue. We have followed this suggestion and introduced a few sentences and relevant quotations in the discussion, addressing the issue of the potential transmission of *Asaia* to humans, and the issue of the potentially increased pro-inflammatory activity of the engineered *Asaia*, linked with WSP expression.

Reviewer #3 (Remarks to the Author):

Based on the contribution of immune priming to Wolbachia-mediated pathogen interference, the authors expressed Wolbachia surface protein (WSP), a potential inductor of immunity response, in *Asaia* bacteria, and observed activation of a number of immune antiviral or anti-plasmodia immune genes after introducing the recombinant bacteria (*Asaia*WSP) into mosquito. Furthermore, the development of the heartworm parasite *Dirofilaria immitis* was inhibited by *Asaia*WSP in *Ae. aegypti*. Overall, this is great work with experimental designed logically. It can facilitate not only better understanding of the mechanism of Wolbachia-mediated pathogen interference in mosquito, but also develop a new artificial symbiont for vector disease control. Thus, I support its publication by Communications Biology after

addressing the below comments. In addition, the writing of this manuscript should be largely improved to make it readable before publishing.

We thank the Reviewer for his positive comments. We have thoroughly revised the manuscript, also following the suggestions of Reviewers 1 and 2, and the manuscript has been revised by a native English-speaking colleague.

Fig. S1, e and f should be list together with Fig. S2, a and b in the same panel for easy comparison.

Done

The statistic information for Fig. S3 should be indicated.

Done

Line 166 and 167, if not statistical significantly, Transferrin should not be thought as over-expressed. It may be ok to say a trend in over-expression was observed.

Done: the sentence was reworded as suggested.

Fig. S4, legend should indicate which figures are 24 hs and 48 hr. Fig 4b, it is better to present the same tissue in parallel for comparison. It is not easy to see the positive signal in Crop and Gut for AsaiaWSP treatment as it is presented currently. The results of Fig 4a should be mentioned in the text before Fig 4b, or the figure label should be switched. In Fig 4a, the medium value should be indicated using a horizontal line.

We thank the reviewer for these suggestions. All the figures or legends have been modified as recommended.

Fig 5, what is the goal to include additional control using sugar and kanamycin?

The idea was to define the baseline of L3 development (in terms of abundance and prevalence), in the absence of any stimulus. The cotton pad with sugar should provide this condition. Kanamycin could potentially determine both an immune activation, as well as it could alter the resident microbiota of the mosquito (with indirect effects on immune gene expression). We thus concluded it was necessary to have also this control, in order to discriminate the (possible) effects determined by kanamycin alone, from those determined by the *Asaia* bacteria (that had been administered in through a cotton pad containing sugar and kanamycin).

214-224, please conclude a reduction only when there is statistic difference. How to interpret no difference in both abundance and prevalence between AsaiaWSP, AsaiapHM4 treatment?

We rewrote the paragraphs on the results of the experiments on *D. immitis* and we re-

analyzed the data on abundance values using the William's mean; statistically significant differences have been presented in separate sentences, in order to avoid any confusion about significant vs non-significant results. As for the lack of significant differences in the comparison between the two *Asaia* strains, we would assume that *Asaia*, with its load of LPS and other immune-stimulating molecules, determines some immune activation in mosquitoes, as recently reported by Cappelli et al. 2019 (Front. Genet. 10:836). Indeed, we also observed an overexpression of immune genes in *Ae. aegypti*, after stimulation with *AsaiapHM4*, for example NOXM (Fig. 3 a). This immune stimulation linked to “*Asaia* itself” could be responsible for the partial reduction of L3 development observed after stimulation with *AsaiapHM4*, thus for the lack of significant differences in the comparison with *AsaiaWSP*. We introduced a couple of sentences in discussion, to address this issue, quoting the paper by Cappelli et al. 2019.

73, trough should be through

Corrected

77, ";" should be behind citation.

Corrected

82, “In addition, it is likely not very resistant outside cells”. It is difficult to understand this sentence

We removed the sentence.

100-102, these sentence should belong to discussion

The sentences have been removed from the introduction.

REVIEWERS' COMMENTS:

Reviewer #3 (Remarks to the Author):

All the comments to the last version have been well addressed in this revised version. The manuscript is now written very well, thus is appropriate to be published by *Communications Biology*.